# Comprehensive Analysis of CPA4 as a Poor Prognostic Biomarker Correlated with Immune Cells Infiltration in Bladder Cancer

**DOI:** 10.3390/biology10111143

**Published:** 2021-11-06

**Authors:** Chengcheng Wei, Yuancheng Zhou, Qi Xiong, Ming Xiong, Yaxin Hou, Xiong Yang, Zhaohui Chen

**Affiliations:** 1Department of Urology, Union Hospital, Tongji Medical College, Huazhong University of Science and Technology, Wuhan 430074, China; chengchengwei@hust.edu.cn (C.W.); m202175943@hust.edu.cn (Y.Z.); xiong_ming@hust.edu.cn (M.X.); m201975738@hust.edu.cn (Y.H.); 2Chongqing Key Laboratory of Molecular Oncology and Epigenetics, Chongqing Medical University, Chongqing 400000, China; xiongqi@stu.cqmu.edu.cn

**Keywords:** CPA4, bladder urothelial carcinoma, immune cells, T cell exhaustion, checkpoint

## Abstract

**Simple Summary:**

The overexpression of Carboxypeptidase A4 (CPA4) has been observed in plenty of types of cancer and has been elucidated to promote tumor growth and invasion; however, its role in bladder urothelial carcinoma (BLCA) is still unclear. Therefore, we aimed to show the prognostic role of CPA4 and its relationship with immune infiltrates in BLCA. We confirmed that the overexpression of CPA4 is associated with shorter overall survival, disease-specific survival, progress-free intervals, and higher dead events. Moreover, we found that several infiltrating immune cells (Th1cell, Th2 cell, T cell exhaustion, and Tumor-associated macrophage) were correlated with the expression of CPA4 in bladder cancer using TIMER2 and GEPIA2. In conclusion, CPA4 may be a novel and great prognostic biomarker based on bioinformation analysis in BLCA.

**Abstract:**

Carboxypeptidase A4 (CPA4) has shown the potential to be a biomarker in the early diagnosis of certain cancers. However, no previous research has linked CPA4 to therapeutic or prognostic significance in bladder cancer. Using data from The Cancer Genome Atlas (TCGA) database, we set out to determine the full extent of the link between CPA4 and BLCA. We further analyzed the interacting proteins of CPA4 and infiltrated immune cells via the TIMER2, STRING, and GEPIA2 databases. The expression of CPA4 in tumor and normal tissues was compared using the TCGA + GETx database. The connection between CPA4 expression and clinicopathologic characteristics and overall survival (OS) was investigated using multivariate methods and Kaplan–Meier survival curves. The potential functions and pathways were investigated via gene set enrichment analysis. Furthermore, we analyze the associations between CPA4 expression and infiltrated immune cells with their respective gene marker sets using the ssGSEA, TIMER2, and GEPIA2 databases. Compared with matching normal tissues, human CPA4 was found to be substantially expressed. We confirmed that the overexpression of CPA4 is linked with shorter OS, DSF(Disease-specific survival), PFI(Progression-free interval), and increased diagnostic potential using Kaplan–Meier and ROC analysis. The expression of CPA4 is related to T-bet, IL12RB2, CTLA4, and LAG3, among which T-bet and IL12RB2 are Th1 marker genes while CTLA4 and LAG3 are related to T cell exhaustion, which may be used to guide the application of checkpoint blockade and the adoption of T cell transfer therapy.

## 1. Introduction

Bladder Urothelial Carcinoma (BLCA) is the eighth most prevalent cancer worldwide, with 549,393 new cases reported worldwide in 2018 [1]. Additionally, in the USA alone, there are estimated to be more than 80,000 new cases and 17,000 deaths each year [2]. This disease is particularly heterogeneous [3]. They are classified as high-grade and low-grade diseases based on standardized histomorphological features, as described by the World Health Organization. The depth of an invasion in the bladder wall determines the tumor stage. Approximately 80% of BLCA patients present non-muscle-invasive bladder cancer (NMIBC) at the time of diagnosis, while the remainder present muscle-invasive bladder cancer (MIBC) or even distant metastases [4]. NMIBCs do not normally pose a threat to patient survival and have a much better prognosis due to effective therapeutic options [5]. However, they almost always relapse, and patients need to repeat intravesical treatments, endoscopic evaluations, and biopsies, which may take an extended period of time, resulting in expensive surgical and surveillance management [6,7,8]. MIBCs, on the other hand, are clinically aggressive and can progress rapidly to lymph nodes, brain, lungs, liver, and bone metastases, which are often fatal [3]. However, over the past three decades, clinical management and five-year survival rates have seen few substantial advances [9]. Therefore, it is significant to identify novel biomarkers and molecular targets for advancing the prognosis of BLCA.

Carboxypeptidase A4 (CPA4) is a member of the zinc-containing metallocarboxypeptidase family [10], which could specifically catalyze the peptide bonds released from carboxy-terminal amino acids [11,12]. CPA4 was first discovered when screening for upregulated mRNA during cancer cell differentiation induced by sodium butyrate [13]. From the cellular and biochemical characteristics, CPA4 is secreted from cells in the form of soluble proenzyme (pro-CPA4), which might play a role in creating a tumor microenvironment [10]. Previous studies have demonstrated that CPA4 is closely associated with the aggressiveness, growth, and differentiation in cancer cells [14,15]. However, the underlying mechanism of CPA4 in BLCA remains unclear.

Recently, CPA4 has shown the potential to be a biomarker in the early diagnosis for certain cancers. Sun et al. have reported that the higher expression level of CPA4 in pancreatic cancer tissues and serum is related to poor prognosis and higher aggressiveness [13]. Previously studied showed that upregulated mRNA levels of CPA4 in androgen-independent prostate cancer cells is associated with the Histone Hyperacetylation signaling pathway [16]. In liver cancer and lung cancer, studies have also shown that the higher expression of CPA4 was closely associated with early diagnosis and poor prognosis [13,17]. Despite the potential significance of CPA4 expression in plenty types of cancer, no previous studies have ever shown the expression levels of CPA4 in bladder cancer, especially with regard to its potential therapeautic and prognostic values. Additionally, the correlation with immune infiltrates of CPA4 in BLCA remains to be investigated. Shao et al. demonstrated that CPA4 overexpression promotes the progression of aggressive clinical stage in pancreatic cancer and that the downregulation of CPA4 inhibits non-small-cell lung cancer growth [15,18]. Therefore, we hypothesized that the level of CPA4 is associated with the prognosis and immune cell infiltration in BLCA.

To test this hypothesis, our study evaluated the role of CPA4 on tumorigenesis and clinical significance based on The Cancer Genome Atlas (TCGA). We compared the different expression level of BLCA in age; gender; pathologic T, N, and M stage; pathology; subtype; and OS. In this study, we found that CPA4 is upregulated in BLCA. Significantly, the risk factors of CPA4 upregulation are correlated with poor prognosis. Additionally, the correlation with immune infiltrates of CPA4 for BLCA is also evaluated. Eventually, we link high CPA4 levels and poor prognosis in BLCA.

## 2. Materials and Methods

### 2.1. Data Source

The Cancer Genome Atlas (TCGA) (https://portal.gdc.cancer.gov/, accessed on 7 September 2021) provides 33 types of clinical and pathological information on cancer for scholars and researchers for free [19]. The expression profiles of CPA4 and clinical information of TCGA cancer data were downloaded from the UCSC Xena (https://xenabrowser.net/datapages/, accessed on 7 September 2021) database. The TCGA database is available publicly in open access format and is available where ethical approval and informed consent of the patients were not necessary [20].

### 2.2. CPA4 Methylation Level Analysis

UALCAN (http://ualcan.path.uab.edu/, accessed on 6 September 2021) is a comprehensive, user-friendly, and interactive web resource for analyzing cancer OMICS data and provides graphs and plots depicting expression profiles and patient survival information for protein-coding, miRNA-coding, and lincRNA-coding genes [21]. The UALCAN online tool was utilized to analyze the CPA4 methylation level in BLCA (TCGA data).

### 2.3. Analysis of Differentially Expressed Genes (DEGs)

Through the limma Package by R, patients with different CPA4 expression profiles in the high and low expression groups (HTSeq-TPM) were compared using unpaired Student’s *t*-test to identify the DEGs [22]. A |log2Fold Change| > 2 and BH-adjusted *p*-values < 0.05 were considered the threshold for the DEGs in a Gene Ontology (GO) Enrichment Analysis. Metascape (https://metascape.org, accessed on 7 September 2021) is a tool used for gene annotation and pathway analysis [23]. In this study, Metascape was utilized to analyze the enrichment of CPA4-related DEGs in processes and pathways. A *p*-value < 0.01, a minimum count of 3, and an enrichment factor of > 1.5 were regarded as significant [24].

### 2.4. Gene Set Enrichment Analysis (GSEA)

GSEA was used as a statistical method in order to seek out whether gene exhibits are statistically significant and concordant between two biological states [25]. We used the R package Cluster Profiler to evaluate excessive function and pathway differences between groups with different expression of CPA4 expression [26]. Each analysis of the processes was repeated 1000 times. Adjusted *p*-value < 0.05 and false discovery rate (FDR) < 0.25 were considered statistically significant enrichments [27]. We chose the potential pathway in which FDR < 0.05 with higher NES after analysis.

### 2.5. Comprehensive Analysis of Protein–Protein Interaction

The Search Tool for the Retrieval of Interacting Genes/Proteins (STRING) website (https://string-db.org/, accessed on 7 September 2021) is a database of known and predicted protein–protein interactions that hosts a collection of integrated and consolidated protein–protein interaction data including direct (physical) and indirect (functional) associations [28]. By importing CPA4 into the online tool STRING, protein–protein interaction (PPI) network information was compiled. Confidence scores > 0.4 were considered median significant.

### 2.6. Analysis of the Tumor Immune Estimation Resource (TIMER2)

The Tumor Immune Estimation Resource (TIMER2) is a comprehensive resource including 32 cancer types and incorporates 10,897 samples from the TCGA database for systematically analysis of immune infiltrates across diverse cancer types (http://cistrome.org/TIMER/, accessed on 7 September 2021) [29]. The TIMER2 database is used to evaluate the correlation of the expression of CPA4 in BLCA patients with the six types of infiltrating immune cells (B cells, dendritic cells, CD4 + T cells, CD8 + T cells, macrophages, and neutrophils) and displays the relationship between the expression of the CPA4 gene and the tumor purity.

### 2.7. Univariate and Multivariate Logistic Regression Analysis

Univariate Cox regression used to calculate the association between OS and patients’ CPA4 expression in two cohorts aims at further researching the effect of CPA4 expression. A multivariate analysis was used to assess if CPA4 is an independent prognostic factor for BLCA patient survival. CPA4 is statistically significant in the Cox regression analysis when the *p*-value is less than 0.05 [30].

### 2.8. Identification of CPA4 Coexpression Genes and Construction of a Prognostic Nomogram

cBiopor tal (https://www.cbioportal.org/, accessed on 7 September 2021) (an online tool based on the TCGA database) was used to identify sets of coexpression genes. According to the *p*-value, we select the most relevant genes about CPA4. Then, the clinical factors (T, M, and N stages; radiation therapy; and primary therapy outcome) and the gene expression levels were used to construct a prognostic nomogram to evaluate the probability of 1-, 2-, and 3-year OS for BLCA patients via the R package (https:// cran.r-project.org/web/packages/rms/, accessed on 7 September 2021) [31].

### 2.9. Immune Infiltration Analysis by ssGSEA

Single sample GSEA (ssGSEA) was performed to analyze the state of immune infiltration of BLCA from R package GSVA (version3.6) (http://www.bioconductor.org/packages/release/bioc/html/GSVA.Html, accessed on 8 September 2021), and we quantified the infiltration levels of 24 immune cell types from gene expression profiles in the literature [32]. In order to discover the correlation between CPA4 and the infiltration levels of 24 immune cells, adjusted *p*-values were established by the Spearman and Wilcoxon rank-sum tests.

### 2.10. Analysis of the Gene Expression Profiling Interactive Analysis 2

The Gene Expression Profiling Interactive Analysis2 (GEPIA2) (http://gepia.cancer-pku.cn/index.html, accessed on 7 September 2021) is an updated database used for analyzing the RNA sequencing expression data of 9736 tumors and 8587 normal samples from the TCGA and the GTEx projects, which include 60,498 genes and 198,619 isoforms [33]. GEPIA2 database investigated the expression level of CPA4 with various immune cells’ markers. TIMER2 was used to identify the gene with a significant correlation with CPA4 expression in the GEPIA2 web.

### 2.11. Statistical Analysis

The expression of CPA4 for non-paired and paired samples was analyzed by the Wilcoxon rank-sum test and Wilcoxon signed-rank test, respectively. By using the pROC package, the ROC curve was generated to evaluate the CPA4 expression with diagnostic performance. The relations between the CPA expression and the clinical features were analyzed by the Kruskal–Wallis test, Chi-Squared test, and Wilcoxon signed rank test. The survival curves were generated via the long-rank test for the Kaplan–Meier analysis. *p* < 0.05 was considered statistically significant: * *p* < 0.05, ** *p* < 0.01, and *** *p* < 0.001; R software was used to process all kinds of statistical analyses (Version 4.0.2). In R, we use *p*adj = *p*.adjust (*p*, method = “BH”, *n* = length(*p*)) to correct the *p*-value.

## 3. Results

### 3.1. Characteristics of BLCA Patients

In total, the information for 414 BLCA tumor tissues and 19 normal tissues were collected from the TCGA database including RNA-seq and relative clinical prognostic information in 414 patients. We grouped the BLCA patients into two sets: low (*n* = 207) and high expressions (*n* = 207) of CPA4. The clinical information of BLCA patients includes age, race, gender, pathologic stage, pathologic stage (T, N, or M), pathologic stage, primary therapy outcome, histologic grade, radiation therapy, subtype, smoking status, lymphovascular invasion, and OS event (Table 1).

### 3.2. Tumor Tissues Express Higher CPA4 Than Normal Tissue

The expression of CPA4 in pan-cancer was analyzed between tumor and normal tissues. From the TCGA + GETx database, the expression level of CPA4 in non-matched patients (*p* = 1.6 × 10^−5^) was significantly higher than that in normal people (Figure 1). The analysis of the correlation between CPA4 expression in BLCA patients and relative clinical information shows that a higher DLEU1 expression level is correlated with OS events and the subtype papillary. No statistically significant differences were found between the expression levels of CPA4 in BLCA and age; gender; pathological T, N, or M stages; and pathologic stage.

### 3.3. Impact of High CPA4 Expression on the Detection and Prognosis of BLCA Patients

The expression of CPA4 indicated a significant discriminative power in identifying tumors from normal cells with an AUC value of 0.798 (Figure 2d). The Kaplan–Meier survival analysis showed that BLCA patients with higher CPA4 expressions have shorter overall survival, disease-specific survival, and progress-free intervals (Figure 2a–c). The KM plots show that a higher expression of CPA4 had a worse prognosis than a lower expression. Promoter methylation of CPA4 in the TCGA-BLCA data was significantly lower than that of normal tissues adjacent to cancer in the UALCAN webpage (*p* < 0.001; Figure 2e).

### 3.4. Differentially Expressed Genes and GO Enrichment Analysis in High- and Low-CPA4 Expression Samples

We analyzed the DEGs in altered expressions of CPA4 including in low and high samples to explore the potential mechanisms of CPA4 that promote tumor progression. There were 529 DEGs identified, of which 349 genes were upregulated and 180 were downregulated (|log2(FC)| > 2 and *p*.adj < 0.05). The DEGs’s expression is shown in a heat map and volcano plot (Figure 3) using GO enrichment analysis to predict the co-expression functions in patients with BLCA. The top GO enrichment items in the biological process (BP), molecular function (MF), and cellular component (CC) groups were epidermal cell differentiation, keratinocyte differentiation, keratinization, intermediate filament cytoskeleton, intermediate filament, cornified envelope, endopeptidase inhibitor activity, peptidase inhibitor activity, peptidase inhibitor activity, serine-type endopeptidase inhibitor activity, metabolism of xenobiotics by cytochrome P450, drug metabolism-cytochrome P450, and retinol metabolism (Figure 4a).

### 3.5. Gene Set Enrichment Analysis for CPA4-Related Signaling Pathways

By the enrichment of MSigDB Collection (c2.all.v7.0.symbols.gmt (curated)), we used the GSEA to identify signaling pathways associated with CPA4 between the different expression levels of CPA4 with significant differences (adjusted *p*-value < 0.05 and FDR < 0.25). The eight pathways included the formation of the cornified envelope, keratinization, immunoregulatory interactions between a lymphoid and a non-lymphoid cell, wp hair follicle development cytodifferentiation part 3 of 3, antigen processing and presentation, assembly of collagen fibrils and other multimeric structures, graft versus host disease, and cytokine–cytokine receptor interaction (Figure 4).

### 3.6. CPA4 Expression Predicts Poor Prognosis in Different Cancer Stages

Univariate cox proportional-hazards model analysis showed that high CPA4 expression, high pathologic grade and stage (T, N, and M), and subtype papillary were negative predictors for OS in BLCA patients. Meanwhile, in the multivariate regression analysis, CPA4 expression was an independent factor correlated with OS both in the low-expression set and high-expression set (*p* = 0.003) (Figure 5).

### 3.7. Construction of Nomogram for Predicting OS and Validation by Calibration

We constructed a nomogram for predicting the prognosis of BLCA with relative clinical situation, which integrates the clinical characteristics associated with the survival of BLCA. Based on the multivariate Cox analysis, a nomogram was assigned to the clinical characteristics of a point and the sum of points awarded to each characteristic is a point from 0 to 100. All of the points are accumulated and recorded as the total points. Using the absolute point axis down to the outcome axis, the probability of BLCA survival at 1, 3 and 5 years can be determined (Figure 6a). From the nomogram, the expression of CPA4 contributes many points compared with other relative clinical situations including the T, N, and M stages; radiation therapy; and primary therapy outcome. Meanwhile, the calibration plot indicates great agreement between the predicted and observed values, which are close to the 45-degree line, which is the ideal curve (Figure 6b).

### 3.8. CPA4-Interaction Protein Networks in BLCA Tissue

CPA4-interaction protein networks were constructed to further explore the necessary proteins for metabolism and the molecular mechanism used by STRING. The PPI network of the CPA4 protein showed the relationship of the CPA4 protein in the progression of BLCA. Ten proteins and corresponding gene names were listed with their annotation scores (Figure 7). The top 10 genes included LXN, CMA1, SGCE, TPSAB1, AGBL2, TPSB2, PEG10, GRB10, TSGA13, and MEST, and LXN had the highest score.

### 3.9. Correlation Analysis between CPA4 Expression and Infiltrating Immune Cells

The survival of patients with different cancers including BLCA is associated with the tumor-infiltrating immune cells. From the result, the expression level of CPA4 had significant correlations with CD8+ T cells (r = 0.287, *p* = 2.29 × 10^−8^), B cells (r = 0.218, *p* = 8.65 × 10^−10^), neutrophils (r = 0.196, *p* = 1.76 × 10^−4^), and dendritic cells (r = 0.356, *p* = 2.5 × 10^−12^). *p* < 0.05 was considered significant (Figure 8a). Furthermore, we analyzed 24 immune cells including pDC, NK CD56bright cells, DC, cytotoxic cells, TFH, B cells, CD8 T cells, Th17 cells, Treg, T cells, mast cells, iDC, NK cells, Tem, aDC, neutrophils, Th1 cells, NK CD56dim cells, macrophages, eosinophils, Tgd T helper cells, Th2 cells, and Tcm. We analyzed the correlation between the expression of CPA4 and immune infiltration by ssGSEA using Spearman’s R. From the result, the expression level of CPA4 was negatively correlated with the infiltration levels of NK CD56bright cells (*p* < 0.001) and positively correlated with cytotoxic cells, T cells, NK cells, idc, Tem, Treg, aDC, Neutrophils, NK CD56dim cells, macrophages, Th2 cells, and Th1 cells (Figure 8).

### 3.10. Possible Role of the Expression of CPA4 in Various Infiltrating Immune Cells

We used the TIMER2 and GEPIA2 databases to further identify the possible role of the expression of CPA4 in various infiltrating immune cells including T cells (general), M1/M2 macrophages, tumor-associated macrophages, B cells, neutrophils, monocytes, NK, CD8+ T cells, and functional DCs as well as T cells such as Th1, Th2, Th9, Th17, Th22, Tfh, exhausted T cells, and Treg. From the results, Th1, T cell exhaustion, and TAM sets marking were greatly connected with the expression of CPA4 in BLCA (Table 2).

## 4. Discussion

CPA4 (carboxypeptidase A4) is a member of the metallocarboxypeptidase family and is a zinc-containing exopeptidase that catalyzes the release of carboxy-terminal amino acids [34]. In recent years, CPA4 has shown the potential to be a biomarker in the early diagnosis with clinical benefit for certain cancers. Some studies revealed that CPA4 is connected with various cancer cells in its differentiation and growth, including non-small-cell lung cancer and gastric cancer [35,36]. Furthermore, it is reported that CPA4 is located on chromosome 7q32 in a region linked to prostate cancer aggressiveness [11], and Sun suggested that CPA4 is closely associated with colorectal cancer liver metastasis [37]. Although CPA4 expression has been confirmed to have potential significance in multiple types of cancer, no studies have shown the expression level and clinical significance of CPA4 in BLCA. In this study, based on a pan-cancer analysis, we demonstrated that human CPA4 expression levels were highly expressed in 11 types of cancer with corresponding normal tissues (Figure 1), which are consistent with the findings in the previous study reported by Sun and Handa et al. [17,35,38]. We also confirmed that CPA4 is significantly upregulated in BLCA (Figure 1b). Moreover, a previous study has shown that CPA4 expression was detected specifically in the cytoplasm of cancer tissue cells, and in the CPA4-suppressed triple-negative breast cancer (TNBC), viability, and migration were decreased [38]. It can act as a potential biomarker of poor prognosis in TNBC. It is reported that CPA4 might be used as an independent poor prognostic factor in esophageal squamous cell carcinoma [39]. In our study, the results in BLCA are consistent. However, one trial showed that CPA4 is a protective factor in muscle-invasive bladder cancer, contrary to the role of CPA4 in most cancers [40]. A potential reason for the difference is due to updates in the TCGA database and different objects. We studied BLCA, while that study investigated muscle-invasive bladder cancer. We compared the different expression levels of BLCA with age; gender; T, N, and M stage; pathologic stage; subtype; and OS. Surprisingly, we found that higher dead events, higher pathologic stages, and the subtype non-papillary were associated with higher expressions of CPA4 in BLCA, with statistical differences (Figure 2). These findings suggest that CPA4 may be a potential biomarker of poor prognosis in identifying BLCA with poor clinical outcome.

Currently, the function of CPA4 in tumors had not been fully reported. Previous trials suggested that the inhibition of CPA4 could reduce the number of breast cancer cells with stemness properties and may be a potential target for TNBC therapy [41]. The CircCPA4 sponge let-7 regulates the expression of CPA4 and glioma progression [42]. All of these results suggest that CPA4 could be regarded as an emerging target or promising biomarker for cancer therapy. Since the mRNA expression of CPA4 in BLCA was significantly higher than that in normal bladder tissue, we speculated that CPA4 can be regarded as a biomarker to detect BLCA from normal controls. To verify the clinical value of CPA4, an ROC curve analysis was performed to verify the clinical value of CPA4 in the diagnosis of BLCA; our results showed that CPA4 may be a potential diagnostic biomarker between bladder cancer and normal tissues, with an AUC of 0.798 (Figure 2d).

Many studies have shown that CPA4 is a significant biomarker of poor prognosis in lots of cancers and is associated with the upregulation of CPA4 with poor overall survival. In hepatocellular carcinoma, AC10364 inhibited cell proliferation and viability through the abnormal expression of genes including CPA4 associated with tumorigenesis or growth [43]. A paper from Yan suggested that the inhibition of CPA4 might be of great significance for improving early stage non-small cell lung cancer survival after ablation [43]. However, the prognostic value in BLCA of CPA4 has not been investigated. With the increased level of CPA4 related to a higher number of dead events and higher pathologic stages, we speculated that CPA4 is involved in the development of BLCA. In light of the Kaplan–Meier curves, we confirmed that the overexpression of CPA4 is associated with shorter overall survival (OS), disease-specific survival (DSF), and progress-free intervals (PFIs) (Figure 2). Moreover, by univariate and multivariate regression analysis, we found that high CPA4 expression; high pathologic stage; T, N, and M stage; and the subtype papillary were negative predictors for OS in BLCA patients and that CPA4 can be an independent factor correlated with OS (Figure 5). The nomogram more accurately predicted 1-, 3-, and 5-year OS in BLCA patients and could help to screen and determine those high-risk patients (Figure 6).

Through GSEA, CPA4 was found to be involved in epidermal cell differentiation, keratinocyte differentiation, keratinization, etc., indicating CPA4 potentially playing a role in cell metabolism and protein synthesis (Figure 4).

The PPI network of CPA4 protein, which were constructed by STRING, showed the relationship of CPA4 in the progression of BLCA such as LXN, CMA1, SGCE, TPSAB1, etc. (Figure 7). It has been reported that latexin (LXN) can inhibit human CPA4, in which the expression is induced in prostate cancer cells after treatment with histone deacetylase inhibitors [44]. The level of CMA1, a key gene, is significantly correlated with gastric cancer prognosis and infiltration level [45]. SGCE promotes breast cancer stem cell self-renewal, chemoresistance, and metastasis both in vitro and in vivo by stabilizing EGFR [45]. Thus, it is speculated that a high expression of CPA4 may increase the degree of malignancy of tumors through CPA4 interacting proteins, leading to the deterioration of patients’ conditions.

Moreover, CPA4 plays a specific role in immune infiltration in bladder cancer. Compellingly, we unraveled that several infiltrating immune cells (Th1cell, Th2 cell, T cell exhaustion, and TAM) were correlated with the expression of CPA4 in bladder cancer using TIMER2 and GEPIA2. Type 1 T helper (Th1) cells produce interferon-gamma [46] (Figure 8, Table 2). The dual inhibition of STAT1 and STAT3 activation downregulates the expression of PD-L1 in cancer cells [47]. T-cell exhaustion is a state of T-cell dysfunction that occurs in many chronic infections and cancers [48]. Scholars have observed that CTLA4 was identified as a crucial negative regulator of the immune system, which transmits an inhibitory signal [49].

There are some limitations in our study. First, basic experiments are needed to verify the results, which were conducted with online public databases. Second, in vivo/vitro experiments are needed to further investigate the potential mechanism of the effect of CPA4 on immune invasion in BLCA.

## 5. Conclusions

In conclusion, our study first demonstrated that CPA4 expression increased in BLCA, and univariate and multivariate regression analyses and a nomogram were used to prove that increased CPA4 is correlated with shorter overall survival, which means high risk factors in BLCA patients. Moreover, we illustrated that a high level of CPA4 was positively related to a high pathologic grade; high T, N, and M stages; and the subtype papillary. The immune infiltration in the tumor microenvironment has also been shown to be associated with CPA4. Collectively, this study partially unveiled that CPA4 in BLCA could be regarded as a potential biomarker for diagnosis and prognosis and may play a special role in immune infiltration.

## Figures and Tables

**Figure 1 biology-10-01143-f001:**
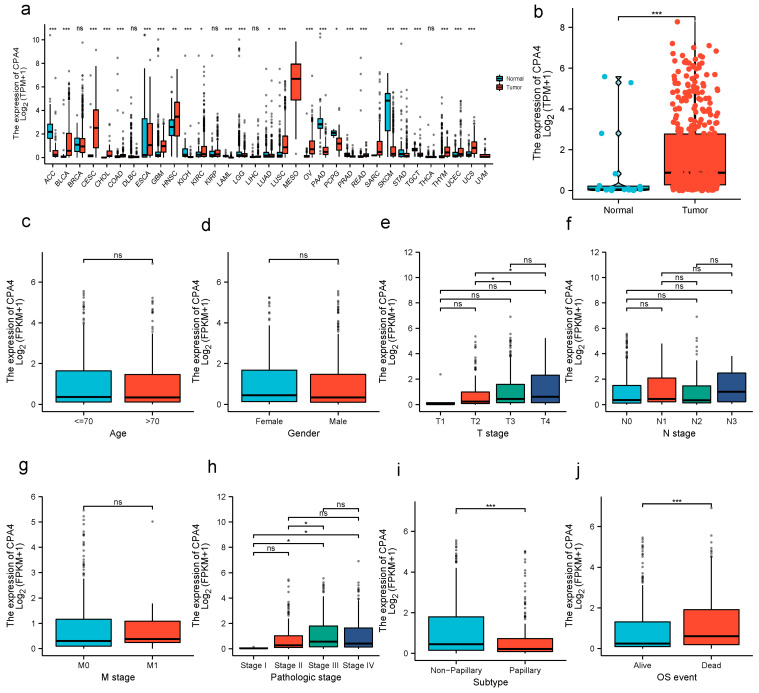
CPA4 expression and clinicopathological features in BLCA. (**a**) human CPA4 expression levels in different cancer tissues and corresponding normal tissues. (**b**) The expression level of CPA4 in BLCA tissue was significantly higher compared with the normal tissues from the TCGA + GTEx database. (**c**–**g**) No statistically significant differences were found between the expression levels of CPA4 in BLCA and age; gender; and pathological T, N, or M stage. (**h**–**j**) High pathologic stage, higher dead event, and nonpapillary were associated with higher expressions of CPA4 in BLCA. * *p* < 0.05; ** *p* < 0.01; *** *p* < 0.001; ns: no significance.

**Figure 2 biology-10-01143-f002:**
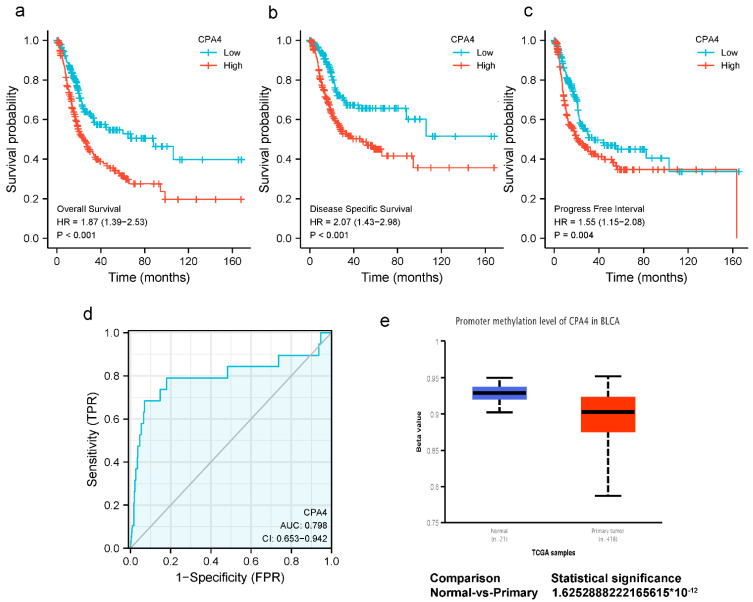
(**a**–**c**) Kaplan–Meier survival curves comparing high and low expressions of CPA4 in BLCA patients. (**a**) overall survival; (**b**) disease-specific survival; (**c**) progression-free interval; (**d**) ROC analysis of CPA4 indicates promising discrimination power between tumor and normal tissues; (**e**) the promoter methylation of CPA4 in tumor tissues (*n* = 418) and normal tissues (*n* = 21) from TCGA-BLCA data.

**Figure 3 biology-10-01143-f003:**
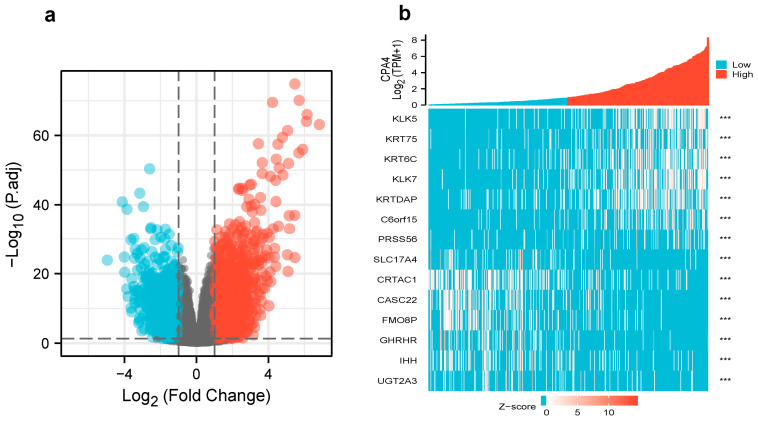
(**a**) Volcano plot of differentially expressed genes (DEGs) connected with the expression of CPA4; (**b**) heatmap of differentially expressed genes (DEGs) connected with the expression of CPA4. *** *p* < 0.001.

**Figure 4 biology-10-01143-f004:**
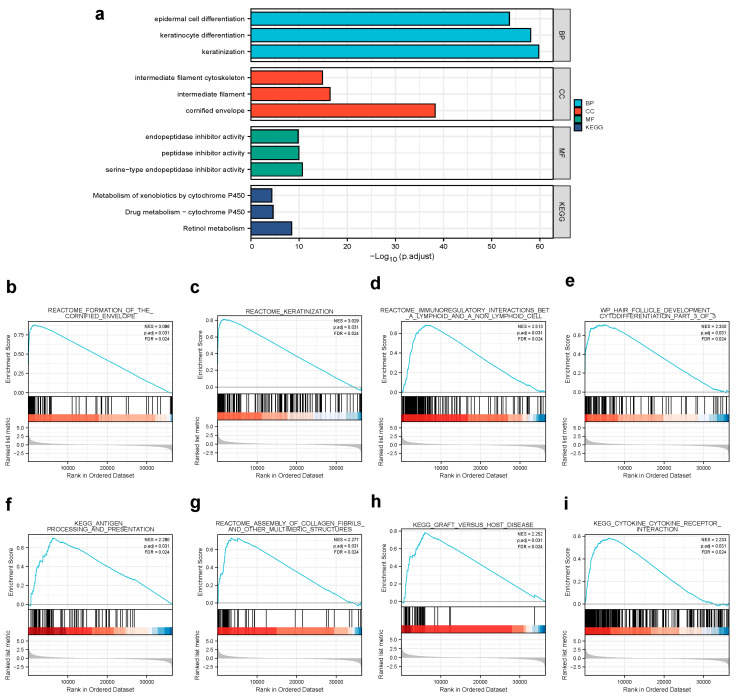
(**a**) GO enrichment analysis of differentially expressed genes (DEGs) in high- and low-CPA4 expression samples; (**b**,**c**) enrichment plots from GSEA. Several pathways were differentially enriched in BLCA patients according to different CPA4 expressions; (**b**) formation of the cornified envelope; (**c**) keratinization; (**d**) immunoregulatory interactions between a lymphoid and a non-lymphoid cell; (**e**) WP hair follicle development cytodifferentiation part 3 of 3; (**f**) antigen processing and presentation; (**g**) assembly of collagen fibrils and other multimeric structures; (**h**) graft versus host disease; (**i**) cytokine–cytokine receptor interaction. ES, enrichment score; NES, normalized enrichment score; ADJ *p*-Val, adjusted *p*-value; FDR, false discovery rate.

**Figure 5 biology-10-01143-f005:**
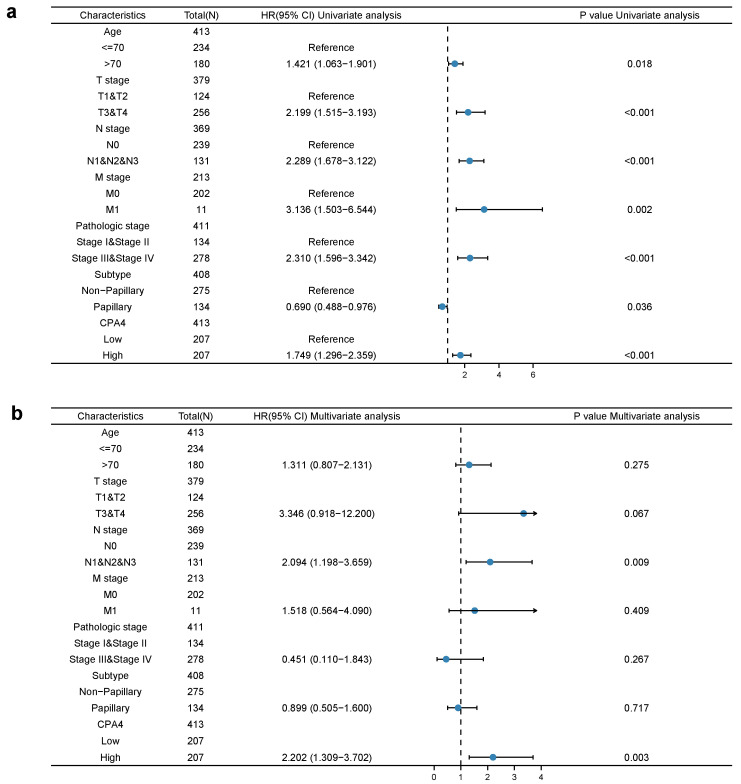
Univariate (**a**) and multivariate (**b**) regression analyses of CPA4 and other clinicopathologic parameters with OS in BLCA patients.

**Figure 6 biology-10-01143-f006:**
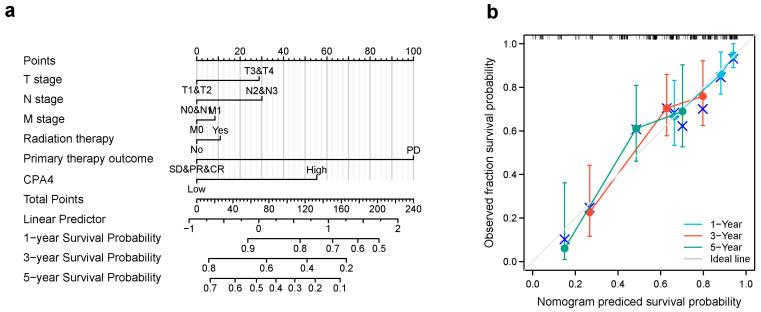
The relationship of CPA4 expression with other clinical factors and overall survival (OS). (**a**) Nomogram for predicting the probability of 1-, 3-, and 5-year OS for BLCA patients; (**b**) calibration plot of the nomogram for predicting the OS likelihood.

**Figure 7 biology-10-01143-f007:**
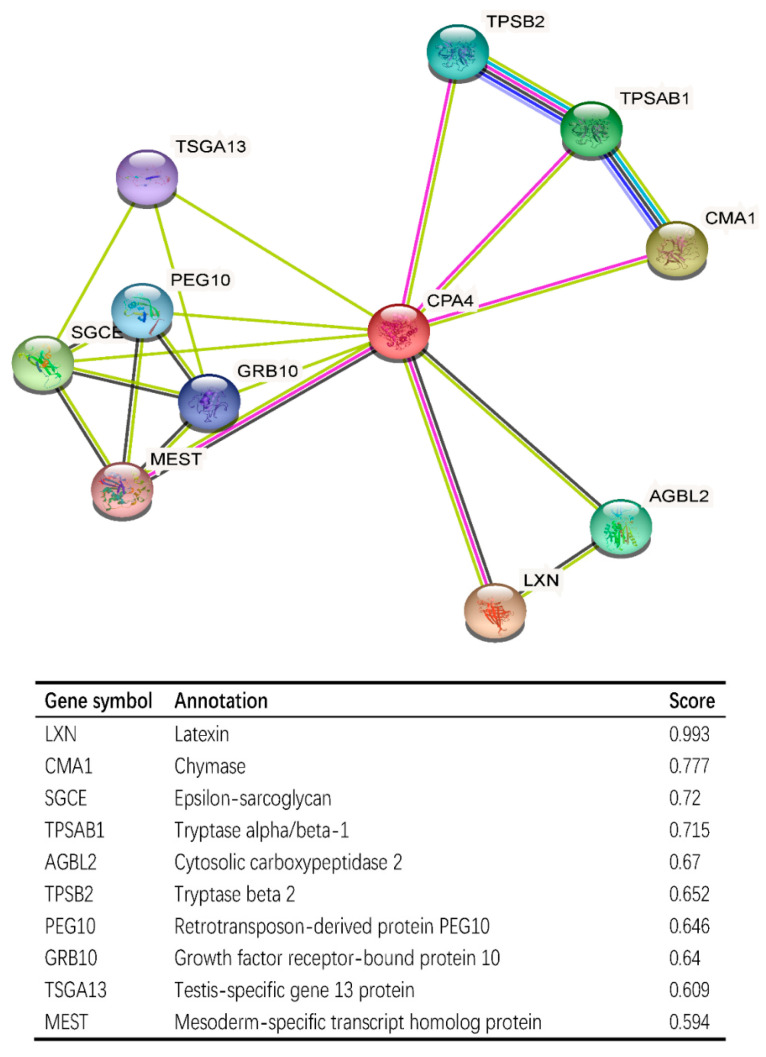
CPA4-interaction proteins in BLCA tissue; annotation of CPA4-interacting proteins and their co-expression scores.

**Figure 8 biology-10-01143-f008:**
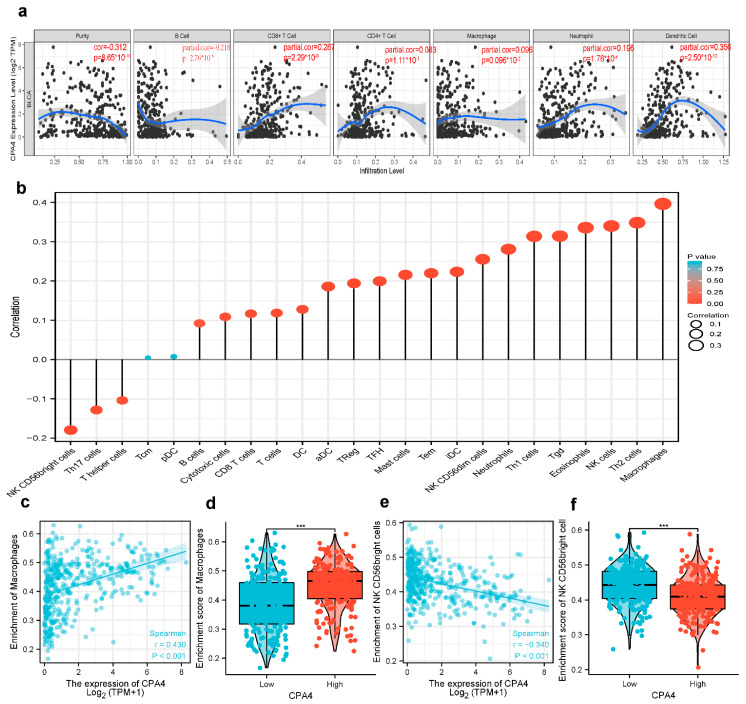
The expression level of CPA4 was related to immune infiltration in the tumor microenvironment. (**a**) Correlation of CPA4 expression with infiltrating immune infiltration in BLCA (**b**) The forest plot shows the correlation between CPA4 expression level and 24 immune cells. The size of the dots indicates the absolute value of Spearman’s R. (**c**,**d**) The Wilcoxon rank sum test was used to analyze the difference in the macrophage cell infiltration levels between the CPA4 high- and low-expression groups; (**e**,**f**) the correlation between CPA4 expression and NK CD56 bright cell infiltration levels. *** *p* < 0.001.

**Table 1 biology-10-01143-t001:** Clinical characteristics of two sets of patients with different expressions of CPA4 from the TCGA dataset.

Characteristic	Low Expression of CPA4	High Expression of CPA4	*p*
*n*	207	207	
Age, *n* (%)			0.921
≤70	116 (28%)	118 (28.5%)	
>70	91 (22%)	89 (21.5%)	
Race, *n* (%)			0.003
Asian	32 (8.1%)	12 (3%)	
Black or African American	8 (2%)	15 (3.8%)	
White	159 (40.1%)	171 (43.1%)	
Gender, *n* (%)			0.372
Female	50 (12.1%)	59 (14.3%)	
Male	157 (37.9%)	148 (35.7%)	
T stage, *n* (%)			0.004
T1	4 (1.1%)	1 (0.3%)	
T2	73 (19.2%)	46 (12.1%)	
T3	89 (23.4%)	107 (28.2%)	
T4	23 (6.1%)	37 (9.7%)	
N stage, *n* (%)			0.494
N0	120 (32.4%)	119 (32.2%)	
N1	18 (4.9%)	28 (7.6%)	
N2	39 (10.5%)	38 (10.3%)	
N3	3 (0.8%)	5 (1.4%)	
M stage, *n* (%)			0.810
M0	109 (51.2%)	93 (43.7%)	
M1	5 (2.3%)	6 (2.8%)	
Pathologic stage, *n* (%)			0.014
Stage I	4 (1%)	0 (0%)	
Stage II	76 (18.4%)	54 (13.1%)	
Stage III	63 (15.3%)	79 (19.2%)	
Stage IV	63 (15.3%)	73 (17.7%)	
Radiation therapy, *n* (%)			0.369
No	181 (46.6%)	186 (47.9%)	
Yes	13 (3.4%)	8 (2.1%)	
Primary therapy outcome, *n* (%)			<0.001
PD	18 (5%)	52 (14.6%)	
SD	14 (3.9%)	17 (4.8%)	
PR	12 (3.4%)	10 (2.8%)	
CR	136 (38.1%)	98 (27.5%)	
Histologic grade, *n* (%)			<0.001
High Grade	186 (45.3%)	204 (49.6%)	
Low Grade	19 (4.6%)	2 (0.5%)	
Lymphovascular invasion, *n* (%)			0.666
No	62 (21.9%)	68 (24%)	
Yes	78 (27.6%)	75 (26.5%)	
Subtype, *n* (%)			0.003
Non-Papillary	124 (30.3%)	151 (36.9%)	
Papillary	82 (20%)	52 (12.7%)	
OS event, *n* (%)			<0.001
Alive	139 (33.6%)	92 (22.2%)	
Dead	68 (16.4%)	115 (27.8%)	
Age, meidan (IQR)	69 (60, 76)	68 (61, 76)	0.990

**Table 2 biology-10-01143-t002:** Correlation analysis between CPA4 and markers of immune cells in BLCA patients found in the TIMER2 and GEPIA2.

Cell Type	Gene Marker	None Cor	*p*	Purity Cor	*p*	Tumor R	*p*	Normal R	*p*
B cell	CD19	−0.042	0.397	−0.138	**	−0.032	0.52	−0.033	0.89
	CD20(KRT20)	−0.314	***	−0.226	***	−0.13	*	−0.18	0.47
	CD38	0.301	***	0.148	**	0.1	*	−0.032	0.9
CD8+ T cell	CD8A	0.267	***	0.12	*	−0.032	0.9	−0.074	0.76
	CD8B	0.15	**	0.018	0.727	0.0031	0.95	−0.096	0.69
Tfh	BCL6	−0.247	***	−0.214	***	−0.14	**	−0.28	0.25
	ICOS	−0.307	***	0.154	**	0.13	**	−0.095	0.7
	CXCR5	0.109	*	−0.095	0.0677	0.075	0.13	0.039	0.87
Th1	T-bet(TBX21)	0.227	***	0.046	0.375	0.2	***	0.041	0.87
	STAT4	0.37	***	0.223	***	0.16	**	0.0068	0.98
	IL12RB2	0.403	***	0.327	***	0.24	***	−0.22	0.37
	WSX1(IL27RA)	0.39	***	0.291	***	0.18	***	0.027	0.91
	STAT1	0.386	***	0.282	***	0.24	***	−0.14	0.56
	IFN-γ(IFNG)	0.278	***	0.161	**	0.13	**	−0.085	0.73
	TNF-α(TNF)	0.287	***	0.194	***	0.098	*	0.27	0.26
Th2	GATA3	−0.484	***	−0.402	***	−0.26	***	−0.26	0.28
	CCR3	0.188	***	0.131	*	0.67	*	−0.14	0.58
	STAT6	−0.228	***	−0.209	***	−0.13	**	−0.32	0.18
	STAT5A	0.004	0.936	−0.158	**	−0.015	0.76	−0.53	*
Th9	TGFBR2	0.087	0.079	−0.014	0.792	0.038	0.45	−0.45	0.056
	IRF4	0.188	***	−0.03	0.571	0.043	0.39	−0.12	0.63
	PU.1(SPI1)	0.356	***	0.181	***	0.15	**	−0.17	0.49
Th17	STAT3	0.325	***	0.232	***	0.15	**	−0.11	0.64
	IL-21R	0.318	***	0.132	*	0.073	0.14	−0.1	0.68
	IL-23R	−0.003	0.945	−0.076	0.143	−0.0048	0.92	−0.019	0.94
	IL-17A	−0.019	0.705	−0.057	0.274	−0.051	0.31	−0.18	0.47
Th22	CCR10	−0.025	0.626	−0.068	0.195	0.029	0.57	−0.34	0.16
	AHR	−0.271	***	−0.195	***	−0.11	*	−0.29	0.23
Treg	FOXP3	0.287	***	0.15	**	0.16	**	0.037	0.88
	CD25(IL2RA)	0.369	***	0.22	***	0.037	0.88	−0.066	0.79
	CCR8	0.218	***	0.083	0.113	0.083	0.094	−0.0059	0.98
T cell exhaustion	PD-1(PDCD1)	0.255	***	0.089	*	0.089	0.073	−0.099	0.69
	CTLA4	0.311	***	0.16	**	0.23	***	−0.11	0.64
	LAG3	0.362	***	0.227	***	0.22	***	−0.19	0.45
	TIM-3(HAVCR2)	0.375	***	0.218	***	0.21	***	−0.097	0.69
Macrophage	CD68	0.316	***	0.193	***	0.14	**	0.49	*
	CD11b(ITGAM)	0.303	***	0.119	*	0.72	*	−0.29	0.23
M1	INOS(NOS2)	−0.033	0.511	−0.092	0.0774	−0.0068	0.89	−0.14	0.57
	IRF5	−0.123	*	−0.116	*	−0.063	0.2	−0.026	0.92
	COX2(PTGS2)	0.209	***	0.164	**	0.057	0.25	−0.24	0.32
M2	CD16	0.408	***	0.273	***	0.17	***	−0.2	0.42
	ARG1	−0.049	0.322	−0.007	0.894	0.076	0.13	0.68	**
	MRC1	0.334	***	0.164	**	0.042	0.4	−0.23	0.34
	MS4A4A	0.353	***	0.199	***	0.12	**	−0.23	0.34
TAM	CCL2	0.26	***	0.113	*	0.022	0.66	−0.12	0.62
	CD80	0.413	***	0.285	***	0.17	***	−0.18	0.46
	CD86	0.396	***	0.244	***	0.17	***	−0.074	0.76
	CCR5	0.29	***	0.101	0.0522	0.13	*	−0.08	0.75
Monocyte	CD14	0.406	***	0.253	***	0.11	*	−0.21	0.38
	CD16(FCGR3B)	0.316	***	0.22	***	0.15	**	−0.073	0.77
	CD115(CSF1R)	0.353	***	0.178	***	0.14	**	−0.28	0.24
Neutrophil	CD66b(CEACAM8)	0.089	0.0721	0.098	0.0609	−0.031	0.53	−0.084	0.73
	CD15(FUT4)	0.141	**	0.047	0.369	0.0041	0.93	−0.33	0.17
	CD11b(ITGAM)	0.303	***	0.119	*	0.018	0.72	−0.29	0.23
Natural killer cell	XCL1	−0.01	0.844	−0.005	0.93	−0.06	0.23	0.13	0.59
	CD7	0.304	***	0.131	*	0.15	**	−0.029	0.91
	KIR3DL1	0.136	**	0.049	0.346	0.075	0.13	0.19	0.44
Dendritic cell	CD1C(BDCA-1)	0.086	0.0823	−0.054	0.305	−0.023	0.65	−0.02	0.93
	CD141(THBD)	0.356	***	0.322	***	0.055	0.27	0.37	0.12
	CD11c(ITGAX)	0.35	***	0.181	***	0.099	*	−0.2	0.41

BLCA, Bladder Urothelial Carcinoma; Tfh, Follicular helper T cell; Th, T helper cell; Treg, Regulatory T cell; TAM, tumor-associated macrophage;.None, correlation without adjustment correlation; Purity, correlation adjusted by purity; Tumor, correlation analysis in the tumor tissue of TCGA; Normal, correlation analysis in normal tissue of TCGA; Cor, R value of Spearman’s correlation * *p* < 0.05; ** *p* < 0.01; *** *p* < 0.001.

## Data Availability

All of the data in this manuscript are available and approved by the institution.

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
