# Peer review of "Comprehensive Analysis of CPA4 as a Poor Prognostic Biomarker Correlated with Immune Cells Infiltration in Bladder Cancer"

_biology, 2021, doi:10.3390/biology10111143_

Round 1

Reviewer 1 Report

The article entitled "Comprehensive analysis of CPA4 as a poor prognostic biomarker correlated with immune cells infiltration in bladder cancer" presents interesting results concern the prognostic role of Carboxypeptidase A4  and its relationship with immune infiltrates in bladder cancer. In general the paper subject is interesting for the readers of Biology MDPI journal. However, in the present form, the article is difficult to analyze and review due to the low quality of the figures, as well as the little discussion regarding the greater amount of data presented. Some improvements are suggested for a next evaluation:

(1) In simple summary, please change BC by BLCA;

(2) Please improve the quality of the pictures in the article, most of them do not have enough resolution for reading and comprehension.

(3) Figure 5 can be changed to a Table(s).

(4) In the discussion topic, figures and tables are not mentioned, making it difficult to understand the data obtained.

(5)The conclusion needs to be rewritten and improved, in its current form it is very reduced and does not present to the reader all the data obtained in this study.

Reviewer 2 Report

In this manuscript, the authors investigated by bioinformatics the potential clinical role of carboxypeptidase A4 (CPA4) in bladder urothelial carcinoma (BLCA). In detail, data from TCGA confirmed the upregulation of CPA4 in BLCA samples and Kaplan-Meier survival analysis showed that BLCA patients with higher CPA4 expression have shorter overall survival, disease-specific survival, and progress-free interval. A potential functional role for CPA4 was obtained after the analysis of differentially expressed genes and GO enrichment analysis in high- and low-CPA4 expression samples. 

Overall, the bioinformatics approach is well-executed. An experimental validations of major results is probably needed. However, my major concern regards the topic of manuscript. In my opinion, this topic is outside the scope of the Journal. In fact, "Cancer" is outside the subject areas of Biology. For this reason, I suggest rejecting the manuscript.

Reviewer 3 Report

Comprehensive analysis of CPA4 as a poor prognostic biomarker correlated with immune cells infiltration in bladder cancer

The manuscript by Wei et .al.  performed a systematic bioinformatics analysis for the investigation of CPA4 in bladder cancer by comparing its expression level in normal tissues and other cancer tissues.  Authors have also analyzed the differentially expressed genes (DEGs), performed functional analysis of genes and analyzed the protein-protein interaction of CPA4 in connection with bladder urothelial carcinoma (BLCA). Furthermore, correlation analysis was performed to evaluate the relationship between CPA4 expression and infiltrating immune cells.  Although the idea is quite interesting, there are several major problems regarding the writing styles of the manuscript as well as for the method and results section:

First of all: the authors hypothesize that they are the first group who shows the importance of CPA4 in bladder cancer. But a short literature review indicates that the following study has already shown the role of CPA4 in bladder cancer.

Link: https://www.frontiersin.org/articles/10.3389/fonc.2019.00856/full

I am wondering therefore  the novelty of this  study. There is a huge discussion need based on the literature review in this manuscript. 

  • Overall, grammatical errors and structure of sentences should be extensively revised, especially in the Results and Discussion section. (Language check). All the other sections also need to be re-written again. Scientific writing is greatly missed. Unless it is corrected, it is not understandable for the scientific community.
  • Design of study and results are interesting, whereas their description supporting the argument could be made stronger with more citations of studies with respect to your results from the study.

Another problem is based on the methods section: The authors use some methodologies without giving any explanation or especially reason, why they prefer such methods. Even worse, they perform some regression analysis without explaining it in methods section.  Furthermore, the definition of p-value thresholds are too artificial and there is statistically no reason to accept something with a FDR<0.25 as significant. In none of the clinical studies or experiments, we can allow such a high error rate. It is not acceptable. Further, i could not find about the p-value correction methods.

Overall the quality of the pictures are too low, the numbers in the pictures cannot be read and for the Table in pages 14 and 15, the number is missing. Further, throughout the manuscript, the abbreviations are not clear stated. Again, the sentences are often not reasonable in the result, discussion and methods sections. For example: in line 29  “Learn from the result, Th1, T cell exhaustion, TAM sets marking were great connected with the expression of CPA4 in BLCA”. What does it mean here, the “great connection”? Please do not construct your sentences with “Learn from ….” 

In total, the authors tried to  do a relatively good job.  The study might have potential to be of interest of the scientific community and anyone interested in bladder cancer. But there are  several huge issues with the manuscript at its current state.

Round 2

Reviewer 2 Report

The manuscript is suitable for the publication

Author Response

Dear reviewer:

Thank you very much for your kind work and consideration on publication of our paper. On behalf of my co-authors, we would like to express our great appreciation to you.

Thank you and best regards.

Yours sincerely,

Zhaohui Chen

Corresponding author: Xiong Yang, Zhaohui Chen

Reviewer 3 Report

I am very glad to see that the authors replied to all of my comments and significantly improved their manuscript. I accept it in present form.

Congratulations!

Author Response

(The authors gave the same response as above.)
